# Assessing the Occurrence and Influence of Cancer Chemotherapy-Related Pharmacogenetic Alleles in the Chilean Population

**DOI:** 10.3390/pharmaceutics16040561

**Published:** 2024-04-19

**Authors:** Gareth I. Owen, Miguel Cordova-Delgado, Bernabé I. Bustos, Leslie C. Cerpa, Pamela Gonzalez, Sebastián Morales-Pison, Benjamín Garcia-Bloj, Marcelo Garrido, Juan Francisco Miquel, Luis A. Quiñones

**Affiliations:** 1Department of Physiology, Faculty of Biological Sciences, Pontificia Universidad Católica de Chile, Santiago 8331150, Chile; cordova.delgado.m@gmail.com (M.C.-D.); pame.gonzalez.h@gmail.com (P.G.); 2Department of Hematology and Oncology, Faculty of Medicine, Pontificia Universidad Católica de Chile, Santiago 7820436, Chile; 3Advanced Center for Chronic Diseases (ACCDiS), Santiago 8330034, Chile; 4Millennium Institute on Immunology and Immunotherapy, Santiago 8331150, Chile; 5Centro de Prevención y Control de Cáncer (CECAN), Santiago 8380453, Chile; 6Faculty of Chemical and Pharmaceutical Sciences, Universidad de Chile, Santiago 8380494, Chile; 7Ken and Ruth Davee Department of Neurology, Simpson Querrey Center for Neurogenetics, Feinberg School of Medicine, Northwestern University, Chicago, IL 60611, USA; bernabe.bustos@northwestern.edu; 8Laboratory of Chemical Carcinogenesis and Pharmacogenetics, Department of Basic and Clinical Oncology, Faculty of Medicine, Universidad de Chile, Santiago 8380494, Chile; leslie.cerpa@uchile.cl; 9Latin American Network for Implementation and Validation of Clinical Pharmacogenomics Guidelines (RELIVAF-CYTED), Santiago 8350499, Chile; 10Centro de Oncología de Precisión (COP), Facultad de Medicina y Ciencias de la Salud, Universidad Mayor, Santiago 7560908, Chile; seba.morales.p@gmail.com (S.M.-P.); benjamin.garcia@umayor.cl (B.G.-B.); marcelo.garrido@umayor.cl (M.G.); 11SAGA, Centro de Estudios Clínicos, Santiago 7610315, Chile; 12Department of Oncología, Clínica Indisa, Santiago 7520440, Chile; 13Department of Gastroenterology, Faculty of Medicine, Pontificia Universidad Católica de Chile, Santiago 8330032, Chile; jmiquelp@uc.cl; 14Department of Pharmaceutical Sciences and Technology, Faculty of Chemical and Pharmaceutical Sciences, University of Chile, Santiago 8380494, Chile

**Keywords:** pharmacogenetic, single-nucleotide polymorphisms, anticancer, Chilean, Latin American, Mapuche-Huilliche, *DPYD*, *TPMT*

## Abstract

Background: Pharmacogenomic knowledge as a biomarker for cancer care has transformed clinical practice, however, as current guidelines are primarily derived from Eurocentric populations, this limits their application in Latin America, particularly among Hispanic or Latino groups. Despite advancements, systemic chemotherapy still poses challenges in drug toxicity and suboptimal response. This study explores pharmacogenetic markers related to anticancer drugs in a Chilean cohort, filling a gap in Latin American research. Notably, the influence of native South American Mapuche-Huilliche ancestry. Methods: To explore pharmacogenetic markers related to anticancer drugs, we utilized an ethnically Admixed Chilean genome-wide association studies (GWAS) dataset of 1095 unrelated individuals. Pharmacogenomic markers were selected from PharmGKB, totaling 36 level 1 and 2 evidence single nucleotide polymorphisms (SNPs) and 571 level 3 SNPs. Comparative analyses involved assessing SNP frequencies across diverse populations from the 1000 Genomes Project. Haplotypes were estimated, and linkage disequilibrium was examined. Ancestry-based association analyses explored relationships between SNPs and Mapuche-Huilliche and European ancestries. Chi-square distribution with *p* ≤ 0.05 and Bonferroni’s multiple adjustment tests determined statistical differences between allele frequencies. Results: Our study reveals significant disparities in SNP frequency within the Chilean population. Notably, dihydropyrimidine dehydrogenase (*DPYD*) variants (rs75017182 and rs67376798), linked to an increased risk of severe fluoropyrimidine toxicity, exhibit an exceptionally low frequency (minor allele frequency (MAF) < 0.005). Nudix hydrolase 15 (*NUDT15*) rs116855232, associated with hematological mercaptopurine toxicity, is relatively common (MAF = 0.062), and is further linked to Mapuche-Huilliche ancestry. Thiopurine methyltransferase enzyme (TPMT), implicated in severe toxicity to mercaptopurines, SNPs rs1142345 and rs1800460 of *TMPT* gene demonstrate higher MAFs in Admixed Americans and the Chilean population (MAF range 0.031–0.057). Finally, the variant in the UDP-glucuronosyltransferase 1 gene (*UGT1A1*) rs4148323, correlated with irinotecan neutropenia, exhibits the highest MAF in East Asian (MAF = 0.136) and Chilean (MAF = 0.025) populations, distinguishing them from other investigated populations. Conclusions: This study provides the first comprehensive pharmacogenetic characterization of cancer therapy-related SNPs and highlights significant disparities in SNP frequencies within the Chilean population. Our findings underscore the necessity for inclusive research and personalized therapeutic strategies to ensure the equitable and effective application of precision medicine across diverse global communities.

## 1. Introduction

Precision medicine, which considers individual genetic variations, is now being used in oncology to improve cancer diagnosis and treatment. The field is also exploring the use of genetic information, such as single-nucleotide polymorphisms (SNPs), to personalize therapy and improve clinical outcomes and quality of life for patients [1].

Pharmacogenomics, the study of how genetic variation affects drug response, has been advanced by genome-wide association studies (GWAS), exon, and site-specific sequencing. However, current GWAS information is primarily based on populations of European descent, and the knowledge base for applying pharmacogenomics in Latin America is hindered by the underrepresentation of Hispanic or Latino populations in genetic analysis [2]. Furthermore, many GWAS studies do not adequately report the origin or genetic background of the study population [1,3].

Despite significant advances in cancer treatment, systemic chemotherapy remains the primary pharmacological approach for most cancer types. However, it is widely recognized that many cancer patients still encounter drug toxicity and suboptimal response rates throughout the course of their illness. In light of this, the identification and validation of genetic variants that contribute to variable drug responses and treatment-related toxicities have played a crucial role in the development of pharmacogenomics testing guidelines. Notably, initiatives such as the Dutch Pharmacogenomics Working Group [4], the Canadian Pharmacogenomics Network for Drug Safety (CPNDS) [5], and the Clinical Pharmacogenomic International Consortium (CPIC) [6] have been instrumental in this endeavor. These guidelines aim to enhance the implementation of pharmacogenomics testing, thereby improving the precision and efficacy of cancer treatments for individual patients.

Among the most remarkable drug–gene pairs are Capecitabine/5-Fluorouracil-*DPYD* [7], Irinotecan-*UGT1A1* [8,9], Mercaptopurines-*NUDT15*/*TPMT* [10], and Vincristine-*CEP72* [11]. Moreover, the *NQO1* rs1800566 variant is related to poorer chemotherapy response in breast cancer therapy, while *GSTP1* rs1695 has been associated with both the effectiveness and toxicity of fluorouracil, cyclophosphamide, and epirubicin-based therapies [12].

On the other hand, multiple studies have demonstrated ancestry-specific patterns in these pharmacogenes [13,14,15], therefore, understanding the prevalence of these variants in multiple populations would help to target efforts in the implementation of pharmacogenetic testing with the greatest possible impact. In this context, the origin of the Chilean population is intricately intertwined with the Mapuche and Huilliche Amerindian communities, which has greatly enriched the nation’s cultural heritage. According to demographic data, the ancestry of the Chilean population is likely a blend of European origins (predominantly Spanish, with some Germanic influence) and the indigenous Mapuche-Huilliche heritage. The fusion of European settlers and these native groups has deeply impacted Chilean society, highlighting the importance of acknowledging their lasting influence [16]. Moreover, this intermingling also has implications for potential variations in drug responses.

Herein, we frame and highlight response/toxicity pharmacogenetics markers related to anticancer drugs in a Chilean cohort of 1095 unrelated individuals, a previously under-explored Latin American population, and we highlight the ancestry origin of those variants and their clinical implication in mestizo populations.

## 2. Methods

### 2.1. Study Population

We used Chilean GWAS data gathered from 1095 unrelated individuals and previously described by the authors [17,18]. Briefly, representative individuals of the Chilean population coming from the ANCORA family health centers in Santiago de Chile (Puente Alto, La Florida, and La Pintana) were included in the study. The sample consists of an ethnically admixed group (European-Amerindian) from urban areas that range in age from 20 to 80 and economically stratified as middle–low income. The participants provided informed consent that was approved by the Ethical Committee at the “Pontificia Universidad Católica de Chile” (Approval No. 12-200, 13 September 2012) and in accordance with the guidelines of the National Commission on Science and Technology (CONICYT-Chile) for usage of genetic data for research purposes, including ancestry-based analyses [17].

### 2.2. PharmacoSNPs Selection

We used the well-established Pharmacogenomic Knowledgebase (PharmGKB) [19]. The complete list of variants/drugs was downloaded in November 2020. The list was then filtered according to the “Related Chemicals” criteria, which included the following drugs associated with the treatment of cancer: capecitabine, fluorouracil, erlotinib, gefitinib, mercaptopurine, methotrexate, cyclophosphamide, letrozole, SN-38, irinotecan, anthracyclines, carboplatin, cisplatin, gemcitabine, trastuzumab, vincristine, ABT-751, bleomycin, cytarabine, daptomycin, daunorubicin, docetaxel, doxorubicin, epirubicin, etoposide, everolimus, exemestane, gemtuzumab, idarubicin, imatinib, itopride, lenalidomide, mitotane, oxaliplatin, paclitaxel, pazopanib, pemetrexed, sirolimus, sorafenib, sunitinib, tamoxifen, tegafur, thiotepa, tipifarnib, topotecan, icotinib, lonafarnib, fludarabine, thalidomide, doxorubicinol, leucovorin, raltitrexed, temsirolimus, mitoxantrone, rituximab, vinorelbine. Once the first filtering was carried out, a second filter according to the level of evidence of the variants was performed identifying 44 SNPs within the group of level 1 and 2 of evidence, and 593 SNPs in level 3. Level 4 variants (only in vitro evidence) were excluded from this study. A workflow is shown in the graphical abstract.

### 2.3. Comparative Analysis of Variants and Haplotypes Frequencies across Different Ethnicities

We extracted 36 SNPs within the group of level 1 and 2 of evidence, and 571 SNPs in level 3, using the GWAS data from Chilean individuals (595 unique SNPs). All SNPs selected were tested for Hardy–Weinberg equilibrium using the Hardy–Weinberg R package goodness-of-fit chi-square test (HW Chisq function, “Hardy Weinberg” package v1.4.1). Data from 1000 Genomes Project on human genome sequence variation phase 3 were taken from the last release of the platform [20]. All the 595 pharmacoSNPs from PharmGKB were found to be present in 1000 Genomes Project variant calls. Allele frequencies were extracted from the 5 Super populations included in the 1000 genomes individuals, including Admixed Americans (AMR), Europeans (EUR), South Asians (SAS), East Asians (EAS), and Africans (AFR). Chi-square distribution with *p*-value of ≤0.05 and Bonferroni’s multiple adjustment tests were used to determine statistical differences in the allele frequencies. The haplotype estimation was performed using the UNPHASED software 3.1.5 version [21]. To measure the linkage disequilibrium among *DPYD*, *ABCB1*, and *TMPT* SNPs we used the software HaploView 4.2 version [22].

### 2.4. Ancestry-Based Association Analysis

Mapuche-Huilliche (M-H) and European ancestry proportions for each of the Chilean GWAS individuals were obtained using ADMIXTURE [23]. The M-H proportions were obtained using whole-genome sequencing data from 12 native individuals from the M-H ethnic group [18]. The European proportion was obtained using the EUR superpopulation dataset from the 1000 genomes project [20]. A linear regression was conducted to assess association between the pharmacogene SNP genotypes and the M-H and European ancestry separately, adjusting by age, sex, and 10 principal components.

## 3. Results

### 3.1. Overview of Selected Variants

The genes associated with the 595 unique variants extracted in our study are predominantly linked to pharmacokinetic and pharmacodynamic processes (pharmacogenes). The majority of these pharmacogenes encode proteins intricately involved in drug metabolism and transport, exerting pivotal influences on the absorption, distribution, and excretion of drugs (Figure 1A). Furthermore, a substantial portion of the selected genes demonstrated connections to cellular processes that undergo alterations in tumor cells. These processes encompass critical facets such as the cell cycle, apoptosis, DNA replication, and repair, all of which serve as targets for chemotherapeutic drugs. On the other hand, a significant portion of the identified SNPs were situated within intronic regions, constituting 37% of the total variants. Additionally, 23% of the identified SNPs manifested as missense variants, which led to alterations in the amino acid sequence of the respective proteins (Figure 1B). Overall, the findings showed that the genes selected were predominantly associated with pharmacokinetic and pharmacodynamic processes, as well as their relevance to cellular mechanisms targeted by chemotherapeutic drugs.

### 3.2. Pharmacogene SNPs Frequency Differences among Chilean Population and 1000 Genomes Superpopulations

We conducted a comprehensive analysis by comparing allele frequencies of 36 level 1 and 2 pharmacogene variants between the Chilean population and the five superpopulations defined in the 1000 Genomes Project. Among these, we identified 28 overlapping SNPs (Figure 2). Notably, the genetic profile of the Chilean population closely mirrors that of the previously reported Admixed American population, while showing pronounced distinctions from the pharmacogenetics observed in the African population for most of the studied SNPs. This observation is likely attributed to the historically low representation of individuals with African ancestry in Chile following the colonial era. In terms of the broader Admixed American population, specific SNPs, including *UGT1A1* rs4148323, *NUDT15* rs116855232, *NT5S2* rs11598702, *FCGR3A* rs396991, *ERCC1* rs3212986, *ERCC1* rs11615, *C8orf34* rs1517114, and *CEP72* rs924607, exhibit slightly higher frequencies in the Chilean cohort, while the remaining 20 SNPs display slightly lower frequencies. Noteworthy differences emerged in comparison to the African cohort, particularly in SNPs such as *ABCB1* rs1045642, *CEP72* rs924607, *ERCC1* rs11615, *NQ01* rs1800566, *NT5C2* rs11598702, *NUDT15* rs116855232, *SCL28A3* rs885004, *UGT1A1* rs4148323, and *XRCC1* rs25487, where the Chilean cohort showcases significantly elevated frequencies. Conversely, SNPs like *ACYP2* rs1872328, *CBR3* rs1056892, *CYP2A6* rs28399454, *FCGR2A* rs1801274, *SEMA3C* rs7779029, and *SLC28A3* rs7853758 display considerably lower frequencies in the Chilean cohort. In comparison to the European superpopulation, prominent distinctions arise in SNPs such as *ABCB1* rs1045642, *ERCC1* rs11615, *FCGR3A* rs396991, *CYP2A6* rs1801272, *DPYD* rs75017182, and *CBR3* rs1056892, with higher frequencies in the former. Notably, *ERCC1* rs3212986, *GSTP1* rs1695, *NQ01* rs1800566, *SLC28A3* rs7853758, and *UGT1A1* rs4148323 exhibit lower frequencies in the Chilean cohort in relation to this population. In comparison to the East Asian superpopulation, the Chilean cohort demonstrates lower frequencies in SNPs including *CBR3* rs1056892, *CYP2A6* rs28399454, *NUDT15* rs116855232, *SEMA3C* rs7779029, *SLCO1B1* rs11045879, and *UGT1A1* rs4148323. Conversely, higher frequencies are observed in SNPs such as *C8orf34* rs1517114, *CEP72* rs924607, *CYP2A6* rs1801272, *ERCC1* rs11615, *ERCC1* rs3212986, *GSTP1* rs1695, *NT5C2* rs11598702, *SCL28A3* rs7779029 and rs885004, *TMPT* rs1142345, and rs1800460. Comparison with the South Asian superpopulation reveals lower frequencies in the Chilean cohort for SNPs *ABCB1* rs1045642, *CBR3* rs1056892, *CYP2A6* rs28399454, *ERCC1* rs11615, and *SEMA3C* rs7779029, while higher frequencies are evident for SNPs C8orf34 rs1517114, *CYP2A6* rs1801272, *ERCC1* rs3212986, *GSTP1* rs1695, *NT5C2* rs11598702, *SCL28A3* rs7779029 and rs885004, *SLCO1B1* rs11045879, *TMPT* rs1142345, and rs1800460. Lastly, compared to the African superpopulation, the Chilean cohort exhibits higher frequencies for SNPs *ABCB1* rs1045642, *CEP72* rs924607, *CYP2A6* rs1801272, *ERCC1* rs11615 and rs3212986, *NQ01* rs1800566, *NT5C2* rs11598702, *NUDT15* rs116855232, *SLC28A3* rs7853758, *TMPT* rs1800460, *UGT1A1* rs4148323, and XRCC1 rs25487 (frequencies in Appendix A).

However, as is evident from Figure 2, each population has its own potentially defining characteristic when the frequency of SNPS was compared to the other populations. The African population shows the highest diversity among the SNPs analyzed, with notable increases in the frequency of *SEMA3c* rs7779029*, SLC28A3* rs7853758, *CYP2A6* rs28399454, *ACYP2* rs1872328 and reduced frequency in *ABC1* rs1045642, *CEP72* rs924607 and *ERCC1* rs11615. The European population possessed a notable enrichment in *ERCC1* rs11615 and a reduction compared to other populations examined in *NUDT15* rs116855232. Interestingly, an increase in SNPs *SLCO1B1* rs11045879 and *UGT1A1* rs4148323 and a lower frequency of *C8orf34* rs1517114 was observed in the East Asian population when compared to South Asia and the other populations.

### 3.3. Very Important Pharmacogenes (VIP) SNPs

We then focused on the comparison across populations and multiples level 1 and 2 SNPs within VIP, including *DPYD*, *GSTP1*, *NUDT15*, *TPMT*, and *UGT1A1*. Table 1 provides a comprehensive overview of the allelic frequencies of VIPs in both the Chilean population and five global superpopulations. An initial observation underscores the non-uniform distribution of allelic frequencies among populations across all loci (Table 1). As previously reported, certain VIP SNPs exhibit distinct patterns, as exemplified by the two *DPYD* variants (rs75017182 and rs67376798) associated with heightened susceptibility to severe fluoropyrimidine toxicity. The European population demonstrates the highest frequencies (MAF = 0.024 and 0.007, respectively), contrasting with rarity in Chilean, Admixed American, Asian, and African populations The *GSTP1* variant (rs1695), associated with neurotoxicity and neutropenia induced by platinum compounds, shows higher MAFs in the African, Chilean, and Admixed American populations (0.484, 0.426, and 0.474, respectively). The *NUDT15* gene variant, rs116855232, associated with hematological toxicity from mercaptopurines, has a higher MAF in East Asian (0.094), South Asian (0.070), and Chilean (0.062) populations, but is extremely rare in European and African populations. Conversely, the rs147390019 variant within the same gene remains rare across all populations. In the context of the *TPMT* gene, implicated in severe toxicity to mercaptopurines, SNPs rs1142345 and rs1800460 exhibit higher MAF in Admixed Americans and the Chilean population compared to other groups, except for the rs1142345 whose MAF was 0.066 for the African population. On the contrary, the variant in the *TPMT* gene, rs1800462, is consistently rare across all populations. Finally, the variant in the *UGT1A1* gene, rs4148323, previously associated with severe neutropenia caused by irinotecan, showed a higher MAF in East Asian (0.136) and Chilean (0.025) populations, distinguishing them from other populations under consideration.

### 3.4. Haplotype Analysis of DPYD, ABCB1, and TPMT Pharmacogenes

It has been reported that the haplotype structure of certain pharmacogenes can significantly impact the activity and function of their respective proteins. For example, haplotypes constructed with SNPs rs75017182, rs2297595, and rs1801265 in the *DPYD* gene increased or decreased dihydropyrimidine dehydrogenase (DPD) activity, which impacts the metabolism of fluoropyrimidines and their potential toxic effects [24]. As reported by Hamzic et al., haplotype GCA (H5) is linked to a significant 16.9% reduction in DPD activity, GCG (H3) results in a 9.6% reduction, and GTG (H2) is associated with an 8.6% increase in DPD activity [24]. Table 2 presents the frequency distribution of haplotypes constructed using the aforementioned SNPs in the Chilean population and the five superpopulations. In the Chilean population, the frequencies of haplotypes correlated with lower DPD activity were 0.015 for H5 and 0.03 for H3, while the frequency for H2, linked to higher DPD activity, was notably higher at 0.148. Particularly, no apparent differences were observed between these haplotypes and the Admixed American population (Table 2). When examining global populations, the European population exhibited the highest frequencies for haplotypes linked to lower DPD activity, with values of 0.032 and 0.087 for H5 and H3, respectively. Conversely, the African population displayed the highest frequency for the haplotype associated with increased activity, H2, reaching a value of 0.414 (Table 2).

Additionally, we constructed haplotypes using the SNPs rs1045642 and rs1128503 in the ABCB1 gene, both associated with the toxicity and efficacy of chemotherapeutics such as methotrexate, platinum, and imatinib. Haplotype analysis revealed that frequencies for A-containing haplotypes (considered risk haplotypes) are notably lower in the African population, followed by the Chilean population (Table 3).

Furthermore, using three relevant SNPs (rs1142345, rs1800460, and rs1800462) in the *TPMT* gene associated with decreased enzyme activity, we constructed haplotypes. Interestingly, the risk haplotype associated with lower enzymatic activity (Table 4) is extremely rare in both Asian populations and among Africans. Conversely, its prevalence is more pronounced in the Admixed American population (0.040) and the Chilean population (0.031).

### 3.5. Ancestry-Related Pharmacogenes SNPs in the Chilean Cohort

Previous studies have highlighted the pharmacogenetic implications associated with ancestry in Latin America [15,25]. Given the diverse composition of the Chilean population, characterized by a blend of European and Native Amerindian roots, primarily Mapuche-Huilliche, we aimed to explore the influence of these ancestries on selected pharmacogene SNPs in our study. Our assessment focused on their associations with Mapuche-Huilliche (M-H) and European (EUR) ancestry proportions, leveraging the Chilean GWAS data [17,18]. Briefly, the M-H proportion in the Chilean GWAS data accounts for an average of 38.64% of the total Admixed American ancestry proportion (45.54%), and the EUR proportion for an average of 49.97%, which represents the combined contribution of the five European subpopulations from the 1000 genomes project phase 3. We identified 18 pharmacogene SNPs exhibiting a significant positive association with M-H ancestry after Bonferroni correction (beta > 0, *p* < 0.05; see Table 5 and Appendix A). Simultaneously, we found 41 pharmacogene SNPs demonstrating a robust positive association with EUR ancestry (beta > 0, *p* < 0.05; refer to Appendix A).

Several significant genetic variations related to pharmacokinetics and pharmacodynamics, such as *ABCB1*, *CYP1A1*, *NAT2*, *NUDT15,* and *SLCO1B1* genes, display robust associations with M-H ancestry. Notably, *NUDT15* rs116855232 (LoE = 1A, *p* = 0.00042), a gene responsible for the biotransformation of thiopurines, stands out as highly associated with ancestry M-H (Table 5 and Appendix A).

In the analysis focusing on individuals of European ancestry, we uncovered 41 significant associations among pertinent pharmacogene SNPs. Notable genes include *ABCC1, CYBA, CYP1A1, FOXO1, GSTA1, GSTP1, IRX5, KDR, NALCN, NOS3, OTOS, PARD3B, UBE21*, and *XPC*. Among these, *GSTA1* rs3957357 (*p* = 3.8 × 10^−6^), *FOXO1* rs144991623 (*p* = 0.00055), *NALCN* rs7992226 (*p* = 0.00029), and *PARD3B* rs17626122 (*p* = 0.00017) exhibited the highest statistical significance. These genes encode proteins that play a pivotal role in drug metabolism and response. Notably, as indicated by PharmGKB annotations, *GSTA1* is associated with platinum and cyclophosphamide response (LoE = 3). *FOXO1* is linked to responses involving cyclophosphamide, epirubicin, and fluorouracil (LoE = 3). *NALCN*’s relevance lies in methotrexate treatment, particularly in cases of lymphoblastic leukemia in children (LoE = 3). Additionally, *PARD3B* is associated with responses to fluorouracil and oxaliplatin (e.g., FOLFOX regimen) (LoE = 3) (Appendix A).

## 4. Discussion

The under-representation of genomic diversity within the South American population, specifically among South American Amerindian communities, poses a significant challenge to the effective implementation of pharmacogenomics and precision medicine. This disparity has the potential to exacerbate health inequalities, particularly between nations where the gene pool is predominantly of European descent. Addressing this issue is crucial for fostering equitable advancements in healthcare and reducing disparities in the understanding and application of genomic information. Here, we reported the allelic frequencies of clinically relevant and potential variants in pharmacogenes related to cancer chemotherapy in a Chilean cohort of 1095 unrelated subjects. This knowledge is instrumental in strategically directing efforts towards the implementation of pharmacogenetic testing, ensuring the maximum impact in improving precision medicine across a broad spectrum of communities.

Examining the 28 SNPs classified with a level of evidence 1 or 2 according to pharmaGKB, a notable distinction emerges between the Chilean population and the African and Asian superpopulations (depicted in Figure 2). This divergence aligns with expectations, given the limited influence of these ancestries on the Chilean population. For example, the estimated genetic contribution from African ancestry is approximately 3–4% in Chileans [26,27].

### 4.1. DPYD Gene

To date, the most reliable markers of fluoropyrimidine toxicity are *DPYD**2A (rs3918290), *DPYD*-c.2846A>T (rs67376798), *DPYD*-*Hap*-B3 (rs75017182), and *DPYD**13 (rs55886062). In fact, these are variants that have a well-documented association with severe toxicity associated with fluoropyrimidines, and there is a Clinical Pharmacogenetics Implementation Consortium (CPIC) guideline that recommends avoiding or reducing the dose of fluoropyrimidines if a patient carries any of these variants [7,28]. In this study, we observed that rs67376798 and rs75017182 have an MAF in the Chilean population of <0.5%, similar to the Admixed American population (<0.06%, Table 1). Gonzalez-Covarrubias et al. reported that the MAF for the SNP rs67376798 was 0.01% in mestizo subjects from Mexico [29]. Several studies from Brazil have reported MAF < 1% for these two DPYD variants [30,31,32]. Furthermore, emerging evidence suggests that considering haplotypes can enhance the interpretation of observed phenotypes in dihydropyrimidine dehydrogenase (DPD) deficiency [24,33]. For instance, Hamzic et al. reported a 10% reduction in DPD activity for the haplotype containing the common variants rs2297595 and rs1801265 (H3) [24]. In line with this, Medwid et al. found that patients with both *DPYD* c.85T>C (rs1801265) and c.496A>G (rs2297595) variants face a significantly increased risk of fluoropyrimidine-related toxicity [33]. Our study demonstrates that for this haplotype (H3), the Chilean population has a frequency of 3% (Table 2), lower than the Admixed American population (4.9%) and notably lower than the European population, which exhibits the highest frequency at 8.7%. Interestingly, a recent study suggested that in an Asian population, the four variants in *DPYD* show a weak association with fluoropyrimidine-induced toxicity, indicating a limited role in this particular population [34]. Given the low frequencies of these variants in the Chilean and Admixed American populations, it is imperative to establish the role of these four variants in fluoropyrimidine-induced toxicity in a clinical context.

### 4.2. UGT1A1 Gene

Irinotecan, a topoisomerase I inhibitor, plays a crucial role in cancer treatment, especially when used in combination therapies for advanced or metastatic colorectal cancer [35]. However, the effectiveness of irinotecan is often accompanied by a notable incidence of toxicity, particularly severe neutropenia and diarrhea [36]. The susceptibility to irinotecan-induced toxicity is heightened by genetic variants linked to diminished UGT enzyme activity, such as *UGT1A**28 [37] and *UGT1A1*6* [38]. In our study, we investigated the prevalence of the *UGT1A1*6* variant (rs4148323) in the Chilean population, revealing a frequency of 0.025. Notably, this frequency is the second highest, surpassed only by the East Asian population with a frequency of 0.136 (Table 1). Consequently, owing to the relatively elevated occurrence of the *UGT1A1*6* variant in the Chilean population, it emerges as a pertinent candidate for genetic screening in patients slated to undergo irinotecan-based treatment. Implementing such screening measures aims to identify individuals at risk and proactively manage irinotecan-induced severe toxicities, thereby enhancing the overall safety and efficacy of the treatment.

### 4.3. NUDT15 and TPMT Genes

TPMT and NUDT15 play crucial roles in the inactivation of thiopurines, and individuals carrying loss-of-function variants in these genes face an increased risk of elevated levels of active metabolites. This heightened sensitivity to thiopurines necessitates a dose reduction to manage potential hematopoietic toxicity effectively [39]. Clinically, the *NUDT15* germline variant rs116855232 is employed to assess the use of thiopurines, especially in the context of acute lymphoblastic leukemia [10]. We observed a relatively high frequency in the Chilean cohort 0.062 (Table 1), while this SNP is observed almost absent in European and African populations, reinforcing the previous suggestion that the MAF of *NUDT15* rs116855232 increases in parallel with the contribution of Native ancestry to the population [40]. Notably, within the Chilean cohort, positive associations between *NUDT15* rs116855232 and Mapuche-Huilliche ancestry were identified (Appendix A). Intriguingly, a study focused on Native populations residing in reservation areas in Brazil unveiled the highest range of rs116855232 MAF (ranging from 19.4% to 31.7% in Kaingang and Guarani populations) reported globally for any population [41]. Furthermore, in a recent study involving Chilean subjects with autoimmune diseases, the rs116855232 variant was reported at a frequency of 9.88% [42], even higher than the frequency identified in our study. Therefore, considering the evidence from this study and others mentioned, genetic testing for this variant should be prioritized in patients slated for thiopurine-based treatment.

Thiopurines, such as 6-mercaptopurine (6-MP), play a pivotal role in the treatment of acute lymphoblastic leukemia (ALL), inflammatory bowel disease, and autoimmune conditions [43,44]. TPMT is a key enzyme responsible for the inactivation of 6-MP, and alterations in its sequence constitute crucial pharmacogenetic predictors with practical implications in clinical settings. Variants like *TPMT*2* (rs1800462), ***3*B* (rs1800460), and ***3*C* (rs1142345) potentially encode enzyme variants unable to detoxify thiopurine byproducts, thereby leading to a decrease in enzymatic function and an increase in toxicity against thiopurines [10]. In our study, the rs1800462 variant was rare in the Chilean population, as well as in other populations (see Table 1). On the other hand, the other two variants, rs1800460 and rs1142345, were relatively common in the Chilean population (MAF = 0.037 and 0.031, respectively), albeit less frequent compared to the Admixed American population (Table 1). Previous reports indicated frequencies of 0% and 1% for rs1800460 and rs1142345, respectively, in Chilean children with ALL [45]. In an Ecuadorian population, the frequencies for these variants were reported as 1.2% and 1.5% (rs1800460 and rs1142345, respectively) [46]. Interestingly, a population from Brazilian Amazonia exhibited higher frequencies of 5.6% and 12.7% for rs1800460 and rs1142345, respectively [47]. Moreover, our study delved into the construction of haplotypes containing all three SNPs in the *TPMT* gene to provide a more comprehensive representation of TPMT activity, a factor linked to increased thiopurine toxicity. The presence of the CTC haplotype, which contains the variants rs1800460 and rs1142345 (*TPMT **3*A*), was associated with significant decreases in TPMT enzymatic activity, resulting in toxicity when thiopurine therapy was administered [48]. Notably, this haplotype appears to be relevant to the Chilean and Admixed American populations, given its frequencies of approximately 3% and 4%, respectively (Table 4).

As highlighted by Suarez-Kurtz et al. [40], when considering combined *TPMT/NUDT15* phenotypes, the current CPIC guidelines for adjusting thiopurine drug doses are applicable to 40% of individuals with significant Native American ancestry. In contrast, this recommendation only pertains to 13% of individuals primarily of European descent. These differences underscore the need for future evaluations in the context of thiopurine toxicity for Chilean mestizo populations, given their significance in thiopurine treatment.

### 4.4. ABCB1 Gene

The *ABCB1* gene, encoding the P-glycoprotein transporter (*MDR1/ABCB1*), plays a pivotal role in drug transport and has implications for drug response and cancer risk. Specifically, the *ABCB1* rs1045642 and rs1128503 variants have been linked to alterations in transporter functionality or expression, as well as associated with the risk of certain cancers or drug-induced toxicity [49]. Constructing a haplotype with the *ABCB1* rs1045642 and rs1128503 variants revealed an intriguing finding—the Chilean population exhibited the second lowest frequency for the AA haplotype among the populations analyzed (Table 3). In a study involving Colombian children with acute myeloid leukemia (AML), the combination of *ABCB1* rs1128503, rs2032582, and rs1045642 AA/AA/AA was strongly associated with death after hematopoietic stem cell transplantation, demonstrating an odds ratio of 13.73 [50]. Similarly, a study in Spanish adults with AML, assessing the same three *ABCB1* variants, found that the haplotype containing A alleles was associated with higher post-induction mortality, increased nephrotoxicity, and hepatotoxicity [51]. Consequently, the lower frequency of the AA haplotype in the Chilean population may influence drug responses and toxicity profiles, particularly in the context of AML, potentially impacting treatment outcomes for conditions influenced by *ABCB1* functionality.

While not the primary focus of this study, it is worth noting that specific SNPs, such as *NQO1* rs1800566, exhibit correlations with cancer incidence, highlighting the broader impact of pharmacogenetic variants. In the Chilean population, *NQO1* rs1800566 shows a higher prevalence compared to Admixed American, European, and African populations (Figure 2), resembling patterns observed in East Asians. Although not significantly associated with overall cancer risk, stratified genotyping studies have linked this SNP to reduced incidences of hepatocellular carcinoma, renal cell carcinoma, stomach cancer in one study [52], and cervical cancer in another [53]. In the DNA damage repair gene *XRCC1*, the allelic frequency of the SNP rs25487 is reduced in the Chilean population compared to the Admixed American population (Figure 2), and this SNP has previously been associated with an increased risk of cervical cancer [53]. These findings underscore the intricate interplay between genetic variations, drug response, and cancer susceptibility, necessitating a comprehensive understanding of personalized therapeutic strategies.

## 5. Conclusions

We present data on cancer chemotherapy-related pharmacogene variant frequencies in *DPYD*, *UGT1A1*, *TPMT*, *NUDT15*, and *ABCB1* within the Chilean population, offering insights for the strategic implementation of pharmacogenetic testing. The observed distinctions in SNPs between the Chilean population and other superpopulations align in general with expectations. In the case of *ABCB1* rs1045642 and rs1128503 variants, the identification of a lower frequency of the AA haplotype in the Chilean population has potential implications for drug responses and toxicity, especially in the context of acute myeloid leukemia. Our work reinforces the concept that the *NUDT15* rs116855232, employed to assess the use of thiopurines, increases in parallel with the contribution of Native ancestry [40] with positive associations between this variant and Mapuche-Huilliche ancestry identified. Thus, we suggest that genetic testing for this variant should be prioritized in patients under consideration for thiopurine-based therapies. This recognition underscores the importance of considering diverse ancestral backgrounds in pharmacogenomic research and emphasizes the imperative of promoting equitable healthcare advancements and reducing disparities in the application of genomic information.

## Figures and Tables

**Figure 1 pharmaceutics-16-00561-f001:**
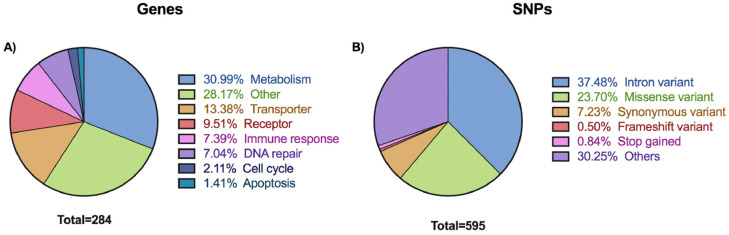
Characteristics of selected 595 unique SNPs and their corresponding 284 genes. (**A**) Distribution of genes regarding this main function, categorized as metabolism, transporter, receptor, immune response, DNA repair, cell cycle, apoptosis, and others. (**B**) Distribution of selected 595 SNPs according to the most severe effect of the variant.

**Figure 2 pharmaceutics-16-00561-f002:**
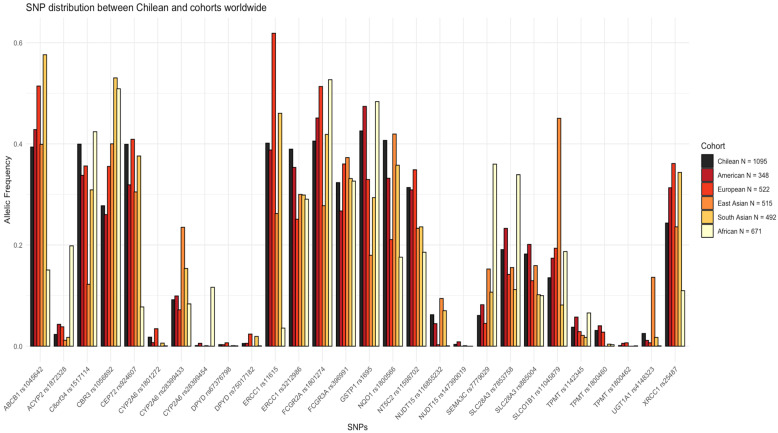
Comparison of 28 SNP, including levels 1 and 2, between the Chilean population and five global superpopulations. Each bar represents the allelic frequency of the respective SNP (X-axis), and each color denotes different populations under comparison.

**Table 1 pharmaceutics-16-00561-t001:** Allelic frequency differences between Chilean and 1000-G super populations in very important pharmacogenes.

SNP ID	Gene	MAF CHI	MAF AMR	*p*	MAF EUR	*p*	MAF EAS	*p*	MAF SAS	*p*	MAF AFR	*p*	LoE	Clinical Annotation	Related Drugs
rs75017182	*DPYD*	0.005	0.006	1	0.024	**1.08 × 10^−^** ^ **5** ^	0	**0.012**	0.019	**0.001**	0.001	**0.023**	1A	Toxicity	Fluoropyrimidines
rs67376798	*DPYD*	0.003	0.003	1	0.007	0.162	0	0.105	0.001	0.448	0.001	0.273	1A	Toxicity	Fluoropyrimidines
rs1695	*GSTP1*	0.426	0.474	**0.025**	0.330	**1.63 × 10^−^** ^ **7** ^	0.180	**3.00 × 10^−^** ^ **45** ^	0.294	**1.11 × 10^−^** ^ **12** ^	0.484	**0.001**	2A	Toxicity	Platinum compounds
rs116855232	*NUDT15*	0.062	0.045	0.093	0.003	**6.24 × 10^−^** ^ **20** ^	0.094	**0.001**	0.070	0.391	0.001	**7.83 × 10^−^** ^ **28** ^	2B	Dosage, Toxicity	Mercaptopurines
rs147390019	*NUDT15*	0.004	0.009	0.117	0.000	0.061	0.001	0.287	0	0.065	0	**0.028**	2B	Dosage, Toxicity	Mercaptopurines
rs1800462	*TPMT*	0.001	0.006	0.062	0.007	**0.016**	0	0.556	0	0.557	0.001	1	1A	Toxicity	Mercaptopurines
rs1142345	*TPMT*	0.037	0.057	**0.030**	0.029	0.219	0.021	**0.018**	0.017	**0.002**	0.066	**2.0 × 10^−3^**	1B	Toxicity	Mercaptopurines
rs1800460	*TPMT*	0.031	0.040	0.274	0.028	0.660	0	**4.42 × 10^−^** ^ **12** ^	0.004	**1.87 × 10^−^** ^ **7** ^	0.003	**2.55 × 10^−^** ^ **10** ^	1B	Toxicity	Mercaptopurines
rs4148323	*UGT1A1*	0.025	0.011	**0.036**	0.007	**2.0 × 10^−3^**	0.136	**6.92 × 10^−^** ^ **32** ^	0.017	0.198	0.001	**7.66 × 10^−^** ^ **11** ^	1B	Toxicity	Irinotecan

MAF = minor allele frequency, LoE = level of evidence, CHI = Chilean, AMR = Admixed Americans, EUR = Europeans, EAS = East Asians, SAS = South Asians, and AFR = Africans. Significant *p* values (<0.05) are in bold and red.

**Table 2 pharmaceutics-16-00561-t002:** Haplotype frequencies of three *DPYD* SNPs in Chilean and 1000-Genome Project Super Populations.

	*DPYD*								
Haplotype	rs75017182	rs2297595	rs1801265	CHI	AMR	EUR	EAS	SAS	AFR
GTA (H1)	G	T	A	0.807	0.763	0.769	0.908	0.732	0.554
** GCA (H5) **	** G **	** C **	** A **	** 0.015 **	** 0.013 **	** 0.032 **	** 0.006 **	** 0.015 **	** 0.006 **
** GTG (H2) **	** G **	** T **	** G **	** 0.148 **	** 0.173 **	** 0.110 **	** 0.072 **	** 0.205 **	** 0.414 **
** GCG (H3) **	** G **	** C **	** G **	** 0.030 **	** 0.049 **	** 0.087 **	** 0.013 **	** 0.046 **	** 0.026 **

CHI = Chilean, AMR = Admixed Americans, EUR = Europeans, EAS = East Asians, SAS = South Asians, and AFR = Africans. Haplotypes associated with lower enzymatic activity are highlighted in red, while the haplotype associated with higher enzymatic activity is highlighted in green in relation to haplotype H1.

**Table 3 pharmaceutics-16-00561-t003:** Haplotype frequencies of 2 *ABCB1* SNPs in Chilean and 1000-Genome Project Super Populations.

	*ABCB1*							
Haplotypes	rs1045642	rs1128503	CHI	AMR	EUR	EAS	SAS	AFR
GG	G	G	0.435	0.507	0.443	0.335	0.327	0.803
AG	A	G	0.089	0.088	0.143	0.039	0.083	0.060
GA	G	A	0.171	0.065	0.042	0.265	0.096	0.047
**AA**	**A**	**A**	**0.305**	**0.340**	**0.371**	**0.359**	**0.492**	**0.091**

CHI = Chilean, AMR = Admixed Americans, EUR = Europeans, EAS = East Asians, SAS = South Asians, and AFR = Africans. Haplotype associated with lower enzymatic activity are highlighted in red.

**Table 4 pharmaceutics-16-00561-t004:** Haplotype frequencies of 3 *TPMT* SNPs in Chilean and 1000-Genome Project Super Populations.

	*TPMT*								
Haplotypes	rs1142345	rs1800460	rs1800462	CHI	AMR	EUR	EAS	SAS	AFR
TCC	T	C	C	0.962	0.942	0.971	0.979	0.983	0.934
CCC	C	C	C	0.006	0.017	0.034	0.021	0.013	0.063
**CTC**	**C**	**T**	**C**	**0.031**	**0.040**	**0.027**	**0.00**	**0.004**	**0.003**

CHI = Chilean, AMR = Admixed Americans, EUR = Europeans, EAS = East Asians, SAS = South Asians, and AFR = Africans. Haplotype associated with lower enzymatic activity are highlighted in red. Haplotype associated with lower enzymatic activity are highlighted in red.

**Table 5 pharmaceutics-16-00561-t005:** Significant pharmacogene SNPs related to Mapuche-Huilliche (M-H) ancestry in the Chilean cohort.

Gene	SNP	*p* *	MA	LoE	Drug	PharmGKB Annotations
*NUDT15*	rs116855232	0.00042	T	1A	Mercaptopurine	Patients with the rs116855232 CC genotype may have increased dose of mercaptopurine as compared to patients with the CT or TT genotype.
*NUDT15*	rs869320766	0.01043	AGGAGTC	1A	Mercaptopurine	The *NUDT15*2* allele is assigned as a no function allele by CPIC. Patients with the *2 allele in combination with a normal or no function allele may be at an increased risk of developing leukopenia or neutropenia when treated with mercaptopurine as compared to patients with two normal function alleles.
*ABCB1*	rs2235047	0.00040	C	3	Anthracyclines	Allele C is associated with increased likelihood of cardiotoxicity when exposed to anthracyclines and related substances in children with neoplasms as compared to allele A.
*BCL2L11*	rs724710	0.03475	T	3	Imatinib	Patients with the TT genotype and cancer may have a diminished response when treated with imatinib as compared to patients with the CC genotype. Other genetic and clinical factors may also influence a patient’s response to imatinib.
*CCAT2*	rs6983267	0.01932	T	3	Platinum Compounds	Patients with the TT genotype and lung cancer may have a decreased response to platinum compounds as compared to patients with the GG and GT genotypes.
*CYP1A1*	rs1048943	0.01654	C	3	Capecitabine/Docetaxel	Women with the GG genotype and breast cancer may have increased progression-free survival time when treated with capecitabine and docetaxel as compared to women with the AA genotype. Other genetic and clinical factors may also influence progression-free survival time.
*DROSHA*	rs639174	0.00544	T	3	Cyclophosphamide/Cytarabine/Daunorubicin/Mercaptopurine/Methotrexate/Prednisone/ Vincristine	Pediatric patients with precursor cell lymphoblastic leukemia–lymphoma and the TT genotype may have an increased risk of drug toxicity when treated with chemotherapy that includes cyclophosphamide, cytarabine, daunorubicin, mercaptopurine, methotrexate, prednisone and vincristine as compared to patients with the CT and CC genotypes. Other clinical and genetic factors may also influence the risk of drug toxicity in pediatric patients with precursor cell lymphoblastic leukemia-lymphoma who are administered chemotherapy.
*EPHX1*	rs1051740	0.04618	C	3	Cisplatin/Cyclophosphamide	Patients with CT genotype and ovarian cancer who are treated with cisplatin and cyclophosphamide may have an increased risk of grade 1–4 nephrotoxicity as compared to patients with CC or TT genotype.
*HOTAIR*	rs7958904	0.03293	C	3	Platinum Compounds	Patients with the CC and CG genotype and lung cancer may have a decreased response to platinum compounds as compared to patients with the GG genotype.
*KCNQ5*	rs9351963	0.00628	C	3	Irinotecan	Patients with the CC genotype and cancer may have an increased risk of diarrhea when treated with irinotecan as compared to patients with the AA or AC genotype.
*NAT2*	rs1799931	0.00280	A	3	Docetaxel/Thalidomide	Patients with AG or GG genotype may have an increased risk of toxicity with docetaxel and thalidomide as compared to patients with the AA genotype.
*NQO1*	rs1800566	0.04034	A	3	Epirubicin/Fluorouracil/Oxaliplatin	Patients with metastatic stomach cancer and the rs1695 AA genotype may have a decreased response to treatment with fluorouracil and oxaliplatin as compared to patients with the GG genotype. Other genetic and clinical factors may also influence response to treatment with fluorouracil and oxaliplatin.
*NT5C3A*	rs3750117	0.00308	A	3	Gemcitabine	Patients with AA or AG genotype and cancer may have increased clearance of gemcitabine, and decreased elimination clearance of dFdU (gemcitabine metabolite) as compared to patients with GG genotype.
*RXRA*	rs2234753	0.00315	G	3	Docetaxel	Patients with nasopharyngeal cancer and the AG genotype who are treated with docetaxel may have more severe anemia as compared to patients with the AA genotype.
*SLC31A1*	rs4978536	0.01282	G	3	Platinum Compounds	Patients with AG or GG genotype and non-small cell lung cancer who are treated with platinum compounds may have an increased severity of thrombocytopenia as compared to patients with AA genotype.
*TP53*	rs4968187	0.00665	T	3	Cyclophosphamide/Epirubicin/Fluorouracil	Patients with breast cancer and the TT genotype may have a decreased risk of developing neutropenia when treated with cyclophosphamide and fluorouracil as compared to patients with the CC or CT genotypes. Other genetic and clinical factors may also affect a patient’s risk of developing neutropenia.
*RAD52*	rs11226	0.00289	A	3	Cisplatin/Cyclophosphamide	Patients with AA or GG genotype and ovarian cancer who are treated with cisplatin and cyclophosphamide may have an increased risk of grade 3–4 neutropenia as compared to patients with the AG genotype.
*SLCO1B1*	rs10841753	0.02520	C	3	Methotrexate	Patients with precursor cell lymphoblastic leukemia-lymphoma and the CC genotype may have increased concentrations of methotrexate as compared to patients with the CT and TT genotypes. There is no association with risk of mucositis or response to methotrexate. Other clinical and genetic factors may also influence concentrations of methotrexate in patients with precursor cell lymphoblastic leukemia-lymphoma.

MA: minor allele; LoE: level of evidence; SNP: single nucleotide polymorphism. *p* *: *p*-value for the association between the pharmacogene SNPs genotypes and the Mapuche-Huilliche ancestry.

## Data Availability

All data generated or analyzed during this study are included in this published article. All allelic frequencies analyzed for the Chilean population are available in the Appendix A files.

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
