# Peer review of "Assessing the Occurrence and Influence of Cancer Chemotherapy-Related Pharmacogenetic Alleles in the Chilean Population"

_pharmaceutics, 2024, doi:10.3390/pharmaceutics16040561_

Round 1

Reviewer 1 Report

Comments and Suggestions for Authors

I thank for the opportunity to review the manuscript titled Assessing the occurence and influence of cancer chemotherapy-related pharmacogenetic alleles in the Chilean population. This is a population study for the assessment of several pharmacogenetic variants related to the oncologic treatment. The manuscript is well written, is clear and easy to follow. However, I have the following comments:

- Please verify that all genes are written in italics. In particular those written in lines 89-92, line 216, line 221, line 260, line 281, line 291, sections 4.1, 4.2 (also alleles), 4.3, 4.4, and lines 496-504. 

- If you have the number of ethical approval, please add it. 

- Delete the dot in line 132. 

- I consider that some examples about the "others" groups included in figure 2 can be briefly mentioned in the text. 

- In the Results section, when the authors described the Figure 2, I think there are some inconsistencies: a) In the figure is stated as NT5C2 and in the text as NT5S2. b) The description of the African population is mentioned twice, one in the line 196, and other in line 2018, are they the same population? c) In lines 192-194, I can see that other SNPs have also slightly higher frequencies in Chilean population: CBR3 rs1056892, CYP2A6 rs1801272, NQO1 rs1800586. d) Lines 203-206 the frequencies were higher in European population, while those SNPs mentioned in lines 205-206 were higher in Chileans. e) The frequencies of CYP2A6 rs28399454 mentioned in line 209 and 214 are really hard to see in the figure 2. f)Line 212 and 217, the rs7779029 do not correspond to SCL28A3 gene. g) Line 222, the frequency of SLC28A3 rs783758 is not higher in Chilean.

- Line 239. The frequency for African population is 0.484 in the table and 0.488 in the text. 

- I highly recommend that the authors describe (in text) the figures and tables in the order that the information is presented (in the figure).

-  Table 1. I suggest to highlight the significant p-values, for a better view of the table.

- Line 269, is DPD activity?

- In all tables, I suggest that the numbers of decimals be constant. 

- Table 5. The NUDT15*2 allele should be written in italics. Please also correct "11harmacogen", which can be found three times. 

- I consider that the Table 6 is too long, and it deviates from the aims of the study. Authors could consider some of this information as a supplementary material or summarize the information presented. 

- Line 419, please verify that is the first time that TPMT is mentioned so the explanation is needed?

- I also consider that the Discussion is long. I think that it should be focused on the relevance of the implementation of some biomarkers according to the frequency or the prevalence of the disease in the Chilean population. 

Reviewer 2 Report

Comments and Suggestions for Authors

Owen et al. investigate pharmacogenetic variation relevant to cancer drugs in Chile with an emphasis on the indigenous Mapuche. 

The study is of importance in Latin America and Chile specifically.

Specific points include:

* How reliable is the estimated Mapuche admixture? What were the ranges? Where other indigenous ethnicities found from Chilane, Bolivian or Argentinian (?) indigenous populations detected?

* Also, what European ancestry were detected? What findings were made related to various European populations? 
* Table 1 should be presented in a larger font, eg going down the page, since it is difficult to read.

* Some of the text can be focused.

Round 2

Reviewer 1 Report

Comments and Suggestions for Authors

The authors considered all my suggestions. I have no further comments.